Combining passive acoustic data from a towed hydrophone array with visual line transect data to estimate abundance and availability bias of sperm whales (Physeter macrocephalus)

Sigourney Douglas B. 1 douglas.sigourney@noaa.gov
DeAngelis Annamaria 2
http://orcid.org/0000-0002-0281-1021 Cholewiak Danielle 2
Palka Debra 2
1 Integrated Statistics , Woods Hole, Massachusetts , United States
2 NOAA Northeast Fisheries Science Center , Woods Hole, Massachusetts , United States
Ward Eric
Electronic publication date: 2023 Sep 21
Publication date: 2023
Volume: 11
Electronic Location ID: e15850
Received 2023 Feb 2; Accepted 2023 Jul 16
Copyright year: 2023
License: This is an open access article, free of all copyright, made available under the Creative Commons Public Domain Dedication. This work may be freely reproduced, distributed, transmitted, modified, built upon, or otherwise used by anyone for any lawful purpose.
License URL: https://creativecommons.org/publicdomain/zero/1.0/

Keywords: Data integration, Multiple data streams, Passive acoustic monitoring, Line transect data, Distance sampling, Mark-recapture, Sperm whales

Funding: National Marine Fisheries Service U.S. Navy N45 Program Bureau of Ocean Energy Management AMAPPS (Atlantic Marine Assessment Program for Protected Species) program M10PG00075 and NEC-11-009 NOAA Fisheries Office of Science and Technology Funding was provided by the National Marine Fisheries Service, the U.S. Navy N45 Program, and the Bureau of Ocean Energy Management. Data were collected as part of the AMAPPS (Atlantic Marine Assessment Program for Protected Species) program under Inter-agency Agreements M10PG00075 and NEC-11-009. Douglas B. Sigourney was supported through funding from the NOAA Fisheries Office of Science and Technology as part of the National Protected Species Toolbox initiative. The funders had no role in study design, data collection and analysis, decision to publish, or preparation of the manuscript.

==============================
Visual line transect (VLT) surveys are central to the monitoring and study of marine mammals. However, for cryptic species such as deep diving cetaceans VLT surveys alone suffer from problems of low sample sizes and availability bias where animals below the surface are not available to be detected. The advent of passive acoustic monitoring (PAM) technology offers important opportunities to observe deep diving cetaceans but statistical challenges remain particularly when trying to integrate VLT and PAM data. Herein, we present a general framework to combine these data streams to estimate abundance when both surveys are conducted simultaneously. Secondarily, our approach can also be used to derive an estimate of availability bias. We outline three methods that vary in complexity and data requirements which are (1) a simple distance sampling (DS) method that treats the two datasets independently (DS-DS Method), (2) a fully integrated approach that applies a capture-mark recapture (CMR) analysis to the PAM data (CMR-DS Method) and (3) a hybrid approach that requires only a subset of the PAM CMR data (Hybrid Method). To evaluate their performance, we use simulations based on known diving and vocalizing behavior of sperm whales (Physeter macrocephalus). As a case study, we applied the Hybrid Method to data from a shipboard survey of sperm whales and compared estimates to a VLT only analysis. Simulation results demonstrated that the CMR-DS Method and Hybrid Method reduced bias by >90% for both abundance and availability bias in comparison to the simpler DS -DS Method. Overall, the CMR-DS Method was the least biased and most precise. For the case study, our application of the Hybrid Method to the sperm whale dataset produced estimates of abundance and availability bias that were comparable to estimates from the VLT only analysis but with considerably higher precision. Integrating multiple sources of data is an important goal with clear benefits. As a step towards that goal we have developed a novel framework. Results from this study are promising although challenges still remain. Future work may focus on applying this method to other deep-diving species and comparing the proposed method to other statistical approaches that aim to combine information from multiple data sources.

Introduction

Worldwide, cetaceans face a growing number of threats presenting challenges for management and conservation (Avila, Kaschner & Dormann, 2018). Effective management relies on accurate and precise estimates of abundance (Hammond et al., 2021). For most large whale species visual line transect (VLT) surveys are a common method to collect information on density and distribution. Generally, to estimate abundance from VLT surveys distance sampling (DS) techniques are applied (Buckland et al., 2001). However, for many species, particularly deep diving cetaceans, VLT surveys alone may not be adequate. Low sample sizes may limit the ability to model ecological relationships (Barkley et al., 2022) and to detect trends in abundance (Taylor et al., 2007). In addition, failure to account for the proportion of animals that are not at the surface, and therefore, not available to be detected can result in considerable bias (Laake et al., 1997, Borchers et al., 2013). This bias, commonly referred to as availability bias (McLaren, 1961; Marsh & Sinclair, 1989), is particularly problematic in deep diving cetaceans and often requires auxiliary data on diving behavior to properly adjust abundance estimates. For these reasons, there is a pressing need to combine VLT surveys with other sources of data, when possible, to achieve accurate and precise estimates of abundance and distribution.

The advent of remote passive acoustic technology to detect and track diving animals offers an opportunity to improve estimates of abundance and estimate availability bias. Passive acoustic monitoring (PAM) can be conducted via a variety of platforms, from stationary recorders deployed at fixed sites on the seafloor, to mobile ones, such as towed hydrophone arrays or gliders. For estimating abundance, towed arrays have proven useful because they can be directly incorporated into a line transect sampling design (Barlow & Taylor, 2005; Gerrodette et al., 2011). However, PAM surveys are also prone to availability bias as animals that are not vocalizing are not available to the PAM survey (Barlow & Taylor, 2005). Studies that utilize both platforms simultaneously by, for example, deploying a towed hydrophone array during VLT surveys have the advantage of potentially minimizing or eliminating this bias as animals are likely be available to at least one platform. Importantly, dual survey designs also have the potential to increase sample size and offer an opportunity to estimate availability bias in situ.

Combing datasets has many potential benefits such as decreasing bias and increasing precision of demographic parameters (Pacifici et al., 2017; Zipkin & Saunders, 2018; Miller et al., 2019; Conn et al., 2022). Although the use of PAM technology is steadily increasing in cetacean research (Marques et al., 2013; Gibb et al., 2019) and several studies have developed methods for analyzing PAM data (Whitehead, 2009; Barlow et al., 2021; Westell et al., 2022), there have been fewer efforts to develop statistical methods for combining PAM data with other data streams such as VLT data. There have been some attempts to compare results from models built with PAM data to models built with VLT data (Barlow & Taylor, 2005; Williamson et al., 2016). In addition, some studies have combined VLT and PAM data with data using existing statistical methods. For example, Gerrodette et al. (2011) used DS methods to combine an estimate of abundance from a VLT survey in deep water habitat with a PAM estimate of abundance from shallow water habitat to get an overall estimate of abundance of vaquitas (Phocoena sinus). Thompson, Brookes & Cordes (2015) used a complementary approach whereby the probability of presence of dolphins was estimated from PAM data and subsequently classified into species group using visual data. Fleming et al. (2018) and Barkley et al. (2022) developed specific protocols for sampling and post-processing data from dual VLT and PAM surveys such the data could be combined into a single species distribution model analysis using standard statistical methods. Despite these novel uses of PAM data, we are not aware of any formal attempts to develop a statistical framework to integrate VLT and PAM data into one estimate of abundance.

Along with increasing precision and accuracy of abundance estimates integrating VLT with PAM data offers an opportunity to estimate and adjust for availability bias in situ. Because of the possibility of large negative bias in abundance estimates, particularly for deep diving cetaceans, there have been numerous attempts to develop methods to account for the diving behavior of animals (Laake et al., 1997; Okamura, 2003; Okamura et al., 2012; Borchers et al., 2013). One common method is to apply a correction factor based on average percent time at the surface (Forcada et al., 2004; Nykänen et al., 2018; Sucunza et al., 2018). However, this method can still result in bias (Borchers et al., 2013). A more sophisticated approach is to integrate information on diving behavior either non-parametrically (Okamura et al., 2012) or through a hidden Markov model approach, but these methods generally require data on the surfacing and diving behavior usually from tagged animals (Borchers et al., 2013; Langrock, Borchers & Skaug, 2013). To date, we are not aware of any studies that attempt to incorporate towed array data into estimates of availability bias. Towed arrays have the advantage of being an independent survey conducted simultaneously with the VLT survey. In addition, unlike tag data where sample sizes are generally low, PAM towed arrays offer the potential for larger samples sizes that could increase precision (e.g., Westell et al., 2022). If the data are sufficiently robust there is also the potential to investigate temporal and spatial variability in availability bias as there may be spatial variability in diving behavior (Watwood et al., 2006).

Although there are many clear benefits to combining VLT and PAM data there are also many challenges. When both surveys are conducted simultaneously one problem is the issue of duplicates. Identifying a sighting as a duplicate is standard protocol for VLT surveys that utilize more than one survey team with the goal of applying mark-recapture distance sampling (MRDS) techniques (Laake & Borchers, 2004; Burt et al., 2014). Although such designs are not uncommon, error associated with duplicate identification is problematic and can lead to bias (Hamilton et al., 2018; Stevenson, Borchers & Fewster, 2019). For surveys that use unmanned aerial platforms it may require novel statistical techniques to objectively define which sightings are duplicates (Stevenson, Borchers & Fewster, 2019; Stevenson, Fewster & Sharma, 2022). With a PAM towed array, this issue has the potential to be particularly challenging because the platforms are surveying animals in different dive states with an unknown probability of animals transitioning among states. As such, a duplicate detection not only depends on the location and timing of the detection but whether an animal transitions among dive states. With regard to location, there is an additional problem of left-right ambiguity with a linear towed array containing two hydrophones as it is difficult to assign a localized detection to one side of the trackline (Barlow & Taylor, 2005; Barkley et al., 2022). Finally, communication among platforms is limited as standard line transect theory requires survey teams to be blind to each to avoid bias (Laake & Borchers, 2004; Burt et al., 2014). These complications make a simple assignment of duplicates challenging just based on the timing and radial distance of a detection.

Sperm whales (Physeter macrocephalus) are an acoustically active species that lend themselves to survey methodologies that use data collected visually or via PAM. Sperm whales have been studied extensively since the 1950s (Worthington & Schevill, 1957; Watkins & Schevill, 1977). When they undergo deep foraging dives, they use echolocation to search for and hone in on their prey (Miller, Johnson & Tyack, 2004). It is the reliability in the pattern of their echolocation clicks and the need to forage regardless of gender or age (barring nursing calves <1 yr old, Tønnesen et al., 2018) that makes sperm whale foraging clicks easy to track while at depth. Foraging clicks (also referred to as ‘usual clicks’ in the literature) have a high source level (236 db; Møhl et al., 2003), can propagate between 5 and 16 km in certain environmental conditions, and have a total vocal active phase of ~35 min, which makes it possible to track diving whales with a hydrophone array towed at the surface. Additionally, the detection range in deep water can exceed that of the known maximum sperm whales’ diving depth of 1,330 m (Madsen, Wahlberg & Møhl, 2002; Mellinger, Thode & Martinez, 2002; Møhl et al., 2003), therefore a sperm whale’s entire search phase is available to be detected (Westell et al., 2022).

In this article we present a statistical method for integrating PAM towed hydrophone array data with VLT data when both surveys are conducted simultaneously. We begin by providing a brief summary of the challenges involved with integrating these two sources of data (see Appendix S1 for a comprehensive list). Next, we describe the statistical framework. Our method combines a capture mark-recapture (CMR) approach with a DS analysis to estimate abundance and surface availability while accounting for transition probabilities among surfacing and diving states. Because processing PAM data can present challenges, we include an alternative approach that requires only a subset of the PAM data be processed into capture histories. We test both methods on simulated data based on diving behavior of sperm whales and compare results to a simpler method that ignores transition probabilities. Finally, we provide a case study from a shipboard survey conducted in the Northwest Atlantic.

Methods

Challenges

There are several challenges to be considered when analyzing PAM data from a towed array data and combining it with VLT data. For example, a submerged whale may not always be available for detection because it could be located in a blind spot where the acoustic signal is blocked from the PAM array or could be in a behavioral state where it is not vocalizing. In addition, there are technical limitations such as the ability to process all the data and confidently assign a series of clicks to an individual whale. In Appendix S1 we provide a detailed outline of the challenges we considered.

Methods to integrate PAM with VLT data

Line transect data are commonly analyzed with DS techniques to estimate detection probability on the trackline where distances (y) of animals or groups of animals are observed. If there is only one survey team the canonical equation can be written as.

(1) p^=∫0w⁡g(y)dyW,

where W is a truncation distance on the trackline and g(y) is a detection function (e.g., half-normal).

One assumption of Eq. (1) is that the detection probability on the trackline (g(0)) is equal to one. Two teams can be introduced to deal with assumptions of detectability on the trackline. In this MRDS design one team can serve as a trial for the other and mark-recapture techniques can be employed to estimate g(0) and adjust the p^. such that.

(2) p^.=g^(0)∫0w⁡g(y)dyW,

A further adjustment can be applied to account for availability bias as

(3) p^=g^(0)∫0w⁡g(y)dyWa^,

where a^ is the surface availability bias correction typically estimated from auxiliary data such as digital recording tags (DTAGs, sensu Johnson & Tyack, 2003). We refer to Eq. (3) as MRDSAV.

Dual line transect surveys with a VLT platform and PAM platform offer the opportunity to record vocalizing animals below the surface while simultaneously observing animals visually above the surface. and therefore, may not require an auxiliary correction factor as in Eq. (3). This data collection design offers the opportunity to estimate the abundance below the surface ( NB) and abundance above the surface ( NS) to calculate an estimate of total abundance ( NT). However, when the two data types are collected simultaneously there is some unknown percentage of animals that may be available to both survey platforms. Ignoring these potential “duplicates” could result in a positive bias in the estimate of abundance. Therefore, estimating NT by combining NB and NS requires an adjustment for the total number of duplicates ( ND) such that

(4) NT=NS+NB−ND

Below we outline a general framework to analyze the acoustic data and describe how the information from the acoustic analysis is integrated with visual line transect data to derive unbiased estimates of abundance and surface availability. We leverage techniques from DS theory (Buckland et al., 2001) and capture mark-recapture (CMR) theory (Royle & Dorazio, 2008). We describe (1) a simple method that ignores duplicates (DS-DS Method), (2) a method for the case when all acoustic detections can be fully annotated (CMR-DS Method) and (3) a method for when only a subset of the acoustic data can be fully annotated (Hybrid Method). For the purpose of this modelling exercise, we assume two behavioral states, a surface state that is available to the VLT platform and a foraging state that is available to the PAM platform. We do not specifically model silent states but will address that assumption within our modelling framework.

DS-DS method

As a default method we first consider a simple approach that ignores duplicates. With this method we simply apply DS methods to both the visual data and the acoustic data and sum the two estimates together for a final abundance estimate (i.e., assume ND = 0). This method is analogous to the method of Gerrodette et al. (2011) where the difference is we use a Bayesian framework to estimate abundance and precision and we apply it to data collected simultaneously rather than data collected from two separate locations. Either Eq. (1) or Eq. (2) can be used to estimate NS. In practice, g(0) is assumed to be 1 for acoustic data so Eq. (1) is used to estimate NB.

CMR-DS method

For this method we assume that all acoustic detections recorded by the passive acoustic towed array from a vocalizing whale can be annotated into a click train and problems with ambiguous click trains (see Appendix S1) are negligible. For each click train, we divide the forward distances of all recorded clicking events into a series of equally spaced distance bins (see Appendix S2). The first bin is the maximum distance that whales are detected by the towed array, similar to the truncation distance in distance sampling (see Fig. 1 for a conceptual diagram).

Figure 1 Conceptual diagram of a foraging whale.

Conceptual diagram illustrating how a click train of a foraging whale is divided into equally spaced intervals where y represents the forward distance, r is the radial distance and x is the perpendicular distance. The red outline details the area of overlap with the visual team where it is assumed that a whale can be detected by either platform depending on its diving state. Note that the intervals continue behind the ship as the acoustic array is capable of picking up signals both in front of and behind the ship.

Because the towed array continuously tracks the position of a clicking whale over a given time period and distance we can treat information collected from the array as CMR data. We adopt a state-space formulation of the Jolly-Seber model using data augmentation (Royle & Dorazio, 2008). The unobserved state of whale i in distance bin j (zi,j) represents its position above or below the surface, where zi,j = 1 indicates that it is below the surface and available to the towed array and zi,j = 0 indicates that is at the surface and not available to the acoustic array. The probability that a whale is available in the first distance bin when it first comes into range of the acoustic array can be modeled as the outcome of a Bernoulli trial as

(5) zi,1∼Bernoulli(γ1)

Whales that are not present below the surface in the first bin (zi,1 = 0) can enter the diving state in subsequent bins given that they were not below the surface in any previous bin. For all other bins the state process is conditional on the previous state such that a whale that has not yet entered the diving state can enter with a probability γj whereas a whale that has already entered the diving sate can remain in that state with probability ϕ. We model this process as

(6) zij∼Bernoulli(ϕzi,j+γj(1−zi,j−1)),

Equation (3) assumes that the probability of remaining in the foraging state is constant regardless of how long a whale is observed. However, the probability of transitioning out of the foraging state is dependent on how long an animal has been in the diving state. Therefore, we model ϕ as a function of time where Timei,j represents the amount of time whale i has already been in the foraging state when it enters bin j which can be written as

(7) Timei,j=Timei,1zi,1+∑j=2J⁡zi,jTj,

where Timei,1 represents the time whale i has already been in the foraging state when it first enters bin 1 and Tj is the amount of time spent in bin j. Because we assume the ship is moving at constant rate then each distance bin represents the same unit of time. Because we do not know how long a whale has been in the foraging state when it first enters the zone of detection we need to assign a prior probability. We use a uniform distribution such that Timei,1~uniform(0, maxTime) where maxTime represents the maximum amount of time a whale can be in the foraging state. We then model the probability of remaining in the foraging phase as

(8) logit(ϕi,j)=α0+αTTimei,j,

To model detection probability we assume the probability of acoustically detecting a whale is a function of its radial distance from the acoustic array. Therefore, we modeled the detection probability (pi,j) of whale i in distance bin j using a logit link function as

(9) logit(pi,j)=β0+β1Ri,j,

where Ri,j is the radial distance of whale i from the acoustic array when it is in bin j and β1 is the effect of radial distance on detection. To calculate Ri,j we use the localized perpendicular distance from the trackline for whale i and the midpoint of forward distance of bin j.

Finally, the likelihood of the observed acoustic data was modeled as the outcome of a Bernoulli trial as

(10) Yij∼Bernoulli(pi,jzi,j),

where Yi,j is an indicator variable that equals 1 when animal i is detected clicking in bin j and 0 otherwise.

By summarizing the unknown states for every whale in each distance bin we estimated a number of derived statistics using data augmentation (Royle, Dorazio & Link, 2007; Royle & Dorazio, 2008). Data augmentation involves increasing the data set of n individuals to size M where M-n represents the number of “pseudo-individuals” with zero observations that have been added to the original dataset of capture histories. Using this approach, we estimate for each distance bin j the number entering the foraging state ( Fj) or number transitioning from the foraging state to the surface state ( Sj) (Table 1). By summing Fj and Sj across bins we also estimate the total number transitioning from the surface state to the foraging state ( FT) and number transitioning from the foraging state to the surface state ( ST). In terms of duplicates, we are only interested in the number of animals that transition within a “zone of overlap” between the VLT and PAM platforms (see Fig. 1), and therefore, we limit the bins in the calculation of FT and ST to this zone. Finally, we estimate the total number of whales that had been below the surface in the foraging state ( NB(CMR)), which is also referred to as the super population using the terminology from Royle & Dorazio (2008) (Table 1). To estimate NB(CMR), FT and ST caution needs to be used to determine what distance bins to include.

Table 1 Parameter and definitions for derived parameters for the CMR-DS Method.

Parameter	Equation	Definition	
Fj	Fj=∑i=1M⁡(1−z1j−1)∗zij	Number entering the foraging state in forward bin j.	
FT	FT=∑j=YMaxDj=YMinD⁡Fj	Total number entering the foraging state in within the window of overlap	
Sj	Sj=∑i=1M⁡zij−1∗(1−zij)	Number of individuals entering the surfacing state in forward bin j	
ST	ST=∑j=YMaxSj=YMinS⁡Sj	Total number entering the surfacing state in within the window of overlap	
NB(CMR)	NB(CMR)=∑i=1M⁡I{∑j=YMaxj=YMin⁡zij>0}	Total number ever in the foraging state over all sampling occasions. Analogous to the superpopulation from a Jolly-Seber analysis.	

Zone of Overlap- To accurately estimate FT and ST it is necessary to define the appropriate zone of overlap between the VLT and PAM platforms by defining the minimum and maximum forward distance bins where duplicates can occur. In addition, for NB(CMR) the zone should be small enough to minimize the possibility of counting repeat divers (see Appendix S1). For example, a whale that transitions to the surfacing state behind the ship will not be available to the VLT platform. Because there are silent states at the beginning and end of a dive, the zone of overlap should take this into account. Therefore, we define Ymax, Ymin,YMaxF, YMinF, YMaxSand YMinSas the maximum and minimum distance bins to include in the calculation of NB(CMR), FT and ST, respectively (see Table 1). We provide a more detailed explanation of defining zones of overlap in Appendix S3.

To integrate the visual and acoustic data we use Eq. (4) to estimate NT. We calculate the total number of duplicates as ND = FT + ST. To estimate NS from the VLT survey either Eqs. (1) or (2) can be used depening on the survey design. We estimate NB(CMR) using the PAM data and the above analysis CMR method.

Finally, because we define availability bias (as) as the proportion of whales that were available to be detected at the surface, it is simply calculated as.

(11) aS=NSNT,

Hybrid method

Problems with ambiguity and processing time may preclude the ability to develop full capture histories for all click train events detected by the towed array. However, there may be enough information from a subset of fully annotated click trains to estimate the transition probabilities among states. These estimates can be used to estimate the proportion of whales below the surface that likely transitioned while in the zone of overlap and therefore are duplicates. Additionally, despite ambiguity it may be possible to localize all acoustically detected whales and assign a perpendicular distance. For this scenario, we developed a hybrid approach between applying DS and CMR to analyze the acoustic data and correct for the number of duplicates.

We premise this method on the assumption that the number of whales entering the foraging state or transitioning to the surface state is approximately constant among all distance bins. As a corollary, the per capita rate (i.e., percent of the total abundance) transitioning to the foraging state (PF) or surface state (PS) within a given bin is also assumed constant. We also assume that all whales detected by the acoustic array can be localized and assigned a perpendicular distance from the trackline, and therefore, we can estimate abundance below the surface using standard DS techniques as previous studies have done (Barlow & Taylor, 2005; Westell et al., 2022). Finally, we assume that the subset of available annotated click trains is representative of all click trains, including those not annotated. Under these assumptions, we model the number of whales transitioning to the foraging state (Fj(H)) or the surfacing state (Sj(H)) as a proportion of the total number below where

(12) Fj(H)=NB(DS)PF

and

(13) Sj(H)=NB(DS)PS

where NB(DS) is the total number below estimated from applying DS to the localized acoustic detections. To calculate FT(H) and ST(H) we multiply Fj(H) and Sj(H) by the total number of distance bins in the zone of overlap, respectively (see Table 2). For example, if we estimate an average of five whales transitioning from the foraging state to the surface forage state ( Fj(H)=5)and there are a total of 10 bins in the zone of overlap then we would estimate a total of 50 duplicates.

Table 2 Parameter and definitions for derived parameters from the Hybrid Method.

Parameter	Equation	Definition	
Fj(H)	Eq. (9) methods	Number entering the foraging state in forward bin j	
Sj(H)	Eq. (10) methods	Number entering the surfacing state in forward bin j	
FT(H)	Fj(H)ZoneF	Total number entering the foraging state within a defined zone of overlap (ZoneF)	
ST(H)	Sj(H)ZoneS	Total number entering the surface state within a defined zone of overlap (ZoneS)	
fj(H)	Same as Fj (see Table 1)	Subset of the number entering the foraging state in bin j estimated from a subset of acoustic capture histories	
sj(H)	Same as Sj (see Table 1)	Subset of the number entering the surfacing state in bin j estimated from a subset of acoustic capture histories	
nB(H)	Same as NB(CMR) (see Table 1)	Subset of the number ever in the foraging state over all sampling occasions estimated from a subset of acoustic capture histories	
PF	PF=f¯(H)nB(H)	Per capita number entering the foraging state per forward bin estimated from the Hybrid Method where f¯(H) is the average number entering the foraging sate per bin	
PS	PS=s¯(H)nB(H)	Per capita number entering the surface state per forward bin estimated from the Hybrid Method where s¯(H) is the average number entering the foraging sate per bin	

To estimate PF and PS we use the subset of click trains that were processed into capture histories. Using this subset of capture histories we estimate the number entering the foraging state fj(H) and surfacing state sj(H) and the superpopulation nB(H) where the lowercase letters indicate that these quantities are estimated from a subset of the PAM data (see Table 2). We provide more detail and an example application of this method in the Case Study section.

Simulations

Our current method makes a number of simplifying assumptions. For example, it assumes a constant probability of entering the foraging state and that remaining in the state is a linear function of time spent foraging. In addition, it does not include individual variation in dive behavior and does not correct for changes in depth. To test the robustness of these methods we designed a simulation framework with the intention of replicating sperm whale diving behavior as closely as possible. We based our simulation on data from Watwood et al. (2006) who used data collected from DTAGs to characterize the dive cycle and vocalizing behavior of sperm whales. We used maximum depth and the percentage of time in various diving states reported in Watwood et al. (2006) to simulate sperm whale diving behavior. Our simulations include an initial silent state at the start of a dive, an acoustically active foraging state and an extended silent state during the ascent. We also included individual variation in dive cycles and maximum depth. We used half-normal functions to simulate the detection process for both the VLT and PAM data. For more details see Appendix S4.

We simulated 100 datasets and applied all three methods to each dataset to estimate abundance and surface availability as well as coefficients of variation (CV) for each estimate. We calculated relative bias as (ϴESTIMATE-ϴTRUE)/ϴTRUE where ϴTRUE represents the true abundance or availability bias and ϴESTIMATE represents the estimate from the model.

Case study with sperm whales

Data collection

We applied this method on sperm whale data collected during the Northeast Fisheries Science Center’s (NEFSC) shipboard survey conducted from 4 July–17 August 2013 off of the northeast US coast (Palka et al., 2017). During this survey, passive acoustic data and visual line transect data were collected simultaneously. A two team visual sampling approach was used such that g(0) could be estimated directly from the visual data. One team was situated in front of the bridge approximately 11.8 m above the water level and the other team was positioned 15.1 m above the water level. Each visual observer searched the waters in front of the ship for groups of whales using high powered 150 powered binoculars. Radial distance and angle were recorded. For each sighting, the data collected to calculate the perpendicular distance was the distance and angle between the observer and center of the group. In addition, each observer estimated the group size of each group. Information on detection covariates collected including Beaufort sea state, visibility sighting cue and magnitude of glare. Further details can be found in Palka et al. (2017).

We conducted PAM surveys concomitantly with VLT surveys using a towed hydrophone array. We typically deployed the towed array in waters of 100 m or greater and approximately 300 m behind the ship during normal visual survey operations. The array was comprised of two oil filled, modular sections, separated by 30 m of cable. There were 8 hydrophones in total, each pair spaced 1 meter apart. Only the last two hydrophones in the array were used as they were the farthest from the ship and therefore recorded less noise from the ship. The hydrophones used in this study were APC International, Ltd. (Mackeyville, PA, model 42-1021; uncalibrated; 36 dB gain added) with a sampling rate of 192 kHz. Palka et al. (2017) provides further details of the array design and subsequent data collection.

Data processing

Recordings were collected and then analyzed using the acoustic software PAMGuard v.1.12.02 (Gillespie et al., 2009). PAMGuard’s click detector was run across all the data on the last hydrophone pair in the array. Click trains that were received on similar bearings were grouped into “events”, where each event in this analysis is defined as a two-dimensional track of a single whale, represented in PAMGuard as a continuous change of received bearings. These events were localized using the Target Motion Analysis 2D Simplex Optimisation algorithm within PAMGuard. Each event was assigned a latitude and longitude position, a timestamp and bearing angle for each click in the event and an ID number. Using this information and the GPS location of the ship, we calculated a radial distance for each annotated click.

We processed a total of 155 acoustic events that could be localized and used in the analysis. Because of the difficulty in annotating all clicks in each event we were only able to fully annotate clicks for a subset of the data. We used a combination of annotated clicks and capture histories at the 1 min level for time periods that had too many whales clicking simultaneously. To develop capture histories we divided the events into one minute time bins and assigned a 1 if any clicks were observed within that time bin and a 0 if otherwise. This approach precluded the need to annotate every click within an event. We converted these time bins into forward distance intervals based on the forward distance of the first recorded click and the constant speed of the ship (see Appendix S2 for more details).

Analysis

Because we only had a subset of click trains in which all of the clicks could be annotated within an event, we could only apply the Hybrid Method to this dataset. To estimate NS from the visual data we used a MRDS approach (Laake & Borchers, 2004). To estimate NB from the acoustic data we used Eq. (1) with a hazard rate function for g(y). For the CMR data we used capture histories developed from the data processing. For a comparison to more traditional methods, we also applied MRDS with an availability bias correction factor to the VLT data where we used an estimate of surface availability from Palka et al. (2017). For a more detailed summary of the sperm whale analysis see Appendix S5.

Model implementation

We performed all analyses in a Bayesian framework using the software R and JAGS (Plummer, 2003). We assumed vague priors for all parameters. We used a burn-in period of 10,000 iterations followed by an additional 15,000 iterations at a thinning rate of 15. Convergence was checked by examining trace plots and calculating Gelman—Rubin statistics, with convergence presumed when R < 1.1 (Brooks & Gelman, 1998).

Results

Simulations

Results from the simulations demonstrated that the DS-DS Method which ignores duplicates had a relatively high positive bias for abundance (+54.4%) and relatively high negative bias for surface availability (−33.0%) (Fig. 2). The CMR-DS Method had the least bias in terms of abundance (–1.2%) and surface availability (−0.2%) where the bias for the Hybrid Method was 3.2% and −3.4% for abundance and surface availability, respectively (Fig. 2). In terms of precision, the CMR-DS Method was more precise than the Hybrid Method. Coefficients of Variation (CVs) for abundance estimates ranged from 0.03 to 0.06 for the CMR-DS Method and 0.05 to 0.15 for the Hybrid Method. For surface availability CVs ranged from 0.03 to 0.07 for the CMR-DS Method and 0.05 to 0.18 for the Hybrid Method.

Figure 2 Relative bias from simulations by method for abundance and availability bias.

Case study

Gelman-Rubin statistics indicated adequate convergence of all parameters (Table S1) and the CMR component of the Hybrid Method provided an adequate fit to the PAM data as evidenced by a Bayesian p-value of 0.28. From the MRDS analysis of the visual data using the Hybrid Method we estimated an effective strip width (ESW) of approximately 3.8 km (CV = 0.21) and a g(0) of 0.80 (CV = 0.10). From the DS analysis of the acoustic data, we estimated an ESW of approximately 4.1 km (CV = 0.08). The estimate of abundance at the surface (NS) was approximately 6% higher than the estimate of abundance below the surface (NB) but with a high overlap in credible intervals (Table 3).

Table 3 Posterior parameter estimates of abundance.

Parameter	Method	Estimate	CV	Upper	Lower	
N S	Hybrid	306	0.38	668	216	
N B	Hybrid	289	0.08	335	249	
N T	Hybrid	426	0.29	814	332	
N T	MRDSAV	516	0.58	1,398	294	
Note:

Posterior parameter estimates of abundance at the surface (NS), abundance below the surface (NB) from the Hybrid Method and total abundance (NT) from both the Hybrid Method and MRDSAV method for sperm whales. The coefficient of variation (CV) and the upper and lower 95% credible interval are provided for each estimate.

The final estimate of abundance from the Hybrid Method combining VLT and PAM survey data was approximately 17% lower than the estimate of abundance from the MRDSAV analysis using only VLT survey data but with a CV that was approximately 50% lower (Table 3). The estimate of surface availability was 0.72 which was similar to the estimate of 0.61 from Palka et al. (2017) with a CV of 0.08 which is lower than the CV of 0.25 reported in Palka et al. (2017).

Discussion

In this study, we explored the challenges of combining PAM towed array data with VLT survey data when the surveys are conducted simultaneously and developed a statistical framework to estimate abundance and surface availability bias. In particular, we focused on the challenge of correcting for duplicate detections. Using simulations and a case study of sperm whales, we explored several methods that range in complexity and data requirements. Our results demonstrate that (1) duplicates can lead to significant bias if ignored (2) modeling transition probabilities can greatly reduce bias in estimates of surface availability and abundance compared to a more simplistic approach and (3) combining these two sources of data can increase precision of estimates of abundance and surface availability.

To evaluate our methodology, we developed simulations based on known diving behavior of sperm whales. Using this simulation design we could assess, firstly, whether or not the process of transitioning among different states in the dive cycle could cause significant bias if ignored, and secondly, if our proposed framework can adequately capture this dynamic and adjust for this bias appropriately. We found significant bias in the DS-DS Method suggesting that the process of transitioning among diving states during a survey can result in a significant number of duplicate detections. The comparatively large reduction in bias exhibited by the CMR-DS Method suggests our framework can capture this dynamic and adjust for this bias. In addition, the Hybrid Method proved to be a viable alternative when the ability to fully annotate all click train events is not feasible. However, it is important to note that the Hybrid Method is less precise than the CMR-DS Method and, the level of precision will be partly influenced by the sample size of click trains that can be fully annotated. Finally, our simulations demonstrated that PAM data has the potential to be used to estimate availability bias.

Simulations can be a powerful tool to test new statistical methods (DiRenzo, Hanks & Miller, 2023). Despite their usefulness, many studies either do not include simulations or use the statistical model as the data generating model in the simulations precluding a true assessment of model misspecification (DiRenzo, Hanks & Miller, 2023). Our goal with this simulation design was to establish the utility of our methodology beyond a theoretical proof of concept. Our simulations were designed to mimic realistic diving behavior in sperm whales to address the challenges we outlined in the Methods. For example, we included silent states, possibility of the same animal being assigned to two different events (double divers) and individual variation in dive behavior (e.g., maximum depth, time at the surface, etc.,). Although our simulations do not capture all the complexity of sperm whale diving and vocalizing behavior, they provide a basis from which to assess the robustness of our methods. The ability of both methods to decrease bias and produce reasonable results lends credence to this framework as a valid statistical tool capable of producing unbiased and precise estimates in the face of complex diving behaviors.

Analysis of the sperm whale dataset with the Hybrid Method produced an estimate of abundance that was similar to an estimate from a MRDSAV analysis using only the VLT survey data but with considerably higher precision. For deep diving cetaceans, abundance estimates from VLT data alone can result in lower precision for two reasons. One reason is simply that there are few detections at the surface resulting in low sample sizes for conventional DS analyses. A second reason is that the correction factor for availability bias is often based on a limited amount of data as well which can result in a large amount of uncertainty being propagated to the final estimate (Sigourney et al., 2020). In addition to low precision, if the data used to estimate availability bias is limited to, for example, a small sample of tagged individuals it may also lead to bias if the sample is not truly representative of average surfacing time. Studies that use only VLT data often need to pool data among species or surveys to have enough data to estimate abundance (Barlow & Forney, 2007). Pooling across species may introduce bias in the detection function and pooling across years may limit the ability to detect trends in abundance. Using the proposed framework we were able to markedly increase the sample of detections by including the PAM data while still taking advantage of the VLT data to derive a more precise estimate of abundance. In our example with field data, we used only 1 year of survey data and were able to achieve a high level of precision. This level of precision can have important management implications, particularly when trying to detect trends (Taylor et al., 2007).

Concomitant with continued technological advancement there has been increasing progress towards developing statistical methods that meld together disparate datasets providing better estimates of demographic parameters to inform management and conservation efforts (Pacifici et al., 2017; Miller et al., 2019). For example, there have been important advancements with species distribution models that use sophisticated integrated modeling techniques (Miller et al., 2019). However, even a simple average among abundance surfaces can yield improved results (Conn et al., 2022). In our example we are adding data sets together but harnessing the information in the PAM data to develop a correction factor for duplicates. By utilizing the PAM data we can increase precision while our method to model transitions can adjust for bias. However, it is also important to note this result may not be realized in all cases. For example, if PAM detections are limited such that there is large variance in both the PAM estimate of abundance and duplicates then applying the proposed methods may decrease precision over a traditional estimate. Moreover, there may be little to be gained from PAM data if, for example, the sample size for the VLT is already large and either an accurate and precise estimate for surface availability bias already exists or availability bias is negligible.

In addition to abundance our approach demonstrates a novel use of PAM data to estimate availability bias. For a comparison, we contrasted our estimate from the Hybrid Method to an estimate reported in Palka et al. (2017). Palka et al. (2017) used DTAG data of sperm whales and the method of Laake et al. (1997) to estimate a correction factor for availability bias. Because DTAG data are limited, the estimate from Palka et al. (2017) was limited to a small sample of whales. We found close agreement between this estimate and the estimate from the Hybrid Method, but the estimate from the Hybrid Method resulted in a CV that was approximately three times lower. By combining information from the PAM and VLT surveys we were able to increase the amount of information to estimate availability bias. This increase in data also allows the opportunity to estimate availability bias at finer scales. Studies of diving behavior of deep diving cetaceans are often based on a small sample size precluding the ability to characterize temporal or spatial scale variation in diving behavior. Watwood et al. (2006) managed to combine information from several tagging studies which allowed them to characterize diving behavior at regional scales. With our method, potentially finer scales could be examined within a region or study area. Ideally, this may allow for more fine-tuned estimates of availability bias to be applied spatially or temporally for more accurate estimates of abundance.

Because of the potential for large bias in abundance estimates, there have been numerous attempts to develop statistical methods to estimate and adjust for availability bias. These methods either develop an estimator to estimate availability bias and apply it as a correction factor (see Laake et al., 1997) or embed information directly into the detection function estimator (Okamura et al., 2012; Borchers et al., 2013; Langrock, Borchers & Skaug, 2013). Most of these approaches require some auxiliary information that is used to correct or adjust the estimator. In our framework, we approach the estimation of availability bias indirectly. By defining availability bias as the proportion of animals that are available at the surface, we can then derive an estimate by combining the estimate of total abundance with an estimate of abundance at the surface from the VLT survey. Our framework does not require auxiliary data on focal follow surveys or tagged animals as input to the model, however, some knowledge of the diving cycle is required to inform the zone of overlap where duplicates can occur. In addition, because the estimate is based on user-defined inputs such as the maximum perpendicular and forward distance it is somewhat bespoke and not directly transferable to other platforms such as aerial surveys.

Limitations and future work

Although initial results are promising, there remain several caveats that could be explored in future extensions of our framework. For example, our methods currently only model transitions into a foraging state. In addition, we do not include an adjustment for depth which may explain the difference between the higher VLT only estimate of abundance and the estimate from the Hybrid Method in our case study. We also do not include group size in the PAM estimate of abundance which may also partly explain the lower estimate of abundance from the Hybrid Method. Another factor that we did not assess in this study is possible error in the location data of localized whales. Finally, we use relatively simple, linear functions to model the PAM data.

Despite these limitations our flexible framework should allow for multiple extensions. For example, information from DTAG data could be used to better model the probability of being in a silent state. Also, new techniques may allow for estimating depth from the PAM towed array such that depth could be included as another source of information (DeAngelis et al., 2017; Barkley, Nosal & Oleson, 2021; Westell et al., 2022). In addition, information on group size is available from the VLT surveys and could be used to inform the PAM data (see Barlow et al. (2021) for an example). Other extensions could also include modelling error in the location data (see Borchers et al., 2010). Finally, more flexible functions could easily be substituted into the CMR component to better capture the dynamics, particularly regarding diving behavior.

Another data limitation in the current analysis is that we only modelled foraging clicks. Typically, non-foraging clicks such as codas occur during ascents and descents of a deep foraging dive (Frantzis & Alexiadou, 2008), therefore the dive would be encapsulated if just marking usual clicks. Slow clicks are also emitted within minutes of usual clicks (Oliveira et al., 2013), thus the individual would be included in the analysis. These clicks could be added to the analysis as they can also be localized and processed (see Barkley et al., 2022). Future modeling attempts, however, should consider using a separate detection function as the detection probability from these events likely differs from foraging clicks.

Along with modifications to the statistical model, further development of the simulation design could be beneficial particularly if our methods are to be applied to other species. With the current simulations we primarily focused on silent states during foraging dives with simulated whales being silent during the initial descent phase and again during the ascent phase. Isojunno & Miller (2015), however, described a more complex array of diving behaviors where they identified six potential behavioral states. Their detailed analysis of these behaviors includes parameter estimates that could be incorporated into future simulations. To extend this simulation design to other species such as beaked whales a different set of behavioral dynamics would also need to be considered. For example, beaked whales engage in shallow non-foraging dives in between longer foraging dives (Tyack et al., 2006) and their echolocation is highly directional (Johnson et al., 2008) which will influence both their overall availability and detectability. A more intricate simulation design which includes several species could potentially be a useful tool for testing and comparing new methods. In addition, acoustic simulations can also play a role in assisting study design (Peel et al., 2014), and therefore, can serve as an important stand-alone tool beyond testing statistical methods.

Conclusions

The use of PAM survey technology is a rapidly growing area in ecological studies creating a fundamental need for analytic tools to analyze and integrate these data with other data streams (Gibb et al., 2019). As more datasets become available it will be necessary to continue to develop a diverse statistical toolbox. As a step towards this end, we focused on the challenges of integrating PAM towed array data with VLT data when the surveys are conducted simultaneously. Our results demonstrate the value of combining these data for abundance estimates and a novel use of PAM data to estimate availability bias for sperm whales. Although results are promising, our methods are not intended as a panacea and clear challenges remain. With the rapid growth in statistical tools, we anticipate continued progress in this arena. Going forward, we recommend rigorous development of simulations to accompany tool development to fully assess the robustness of methods.

Supplemental Information

Supplemental Information 1 List of Challenges.

Click here for additional data file.

Supplemental Information 2 Processing and formatting passive acoustic data.

Click here for additional data file.

Supplemental Information 3 Description of the Zone of Overlap.

Click here for additional data file.

Supplemental Information 4 Description of Simulations.

Click here for additional data file.

Supplemental Information 5 Description of sperm whale analysis with Hybrid Method.

Click here for additional data file.

Supplemental Information 6 Posterior summaries of parameters.

Posterior summaries of parameters from the analysis of the sperm whale data with the Hybrid Method. Gelman-Rubin statistics (G–R) and effective samples sizes (Eff) are included.

Click here for additional data file.

The authors would like to thank the crew of the NOAA ship Henry B. Bigelow, all the scientists that helped collect the data and all acousticians that helped process the sperm whale acoustic data. In addition, we are grateful to the DenMod working group which provided critical feedback on this project.

Additional Information and Declarations

Competing Interests

Author Contributions

Data Availability

Douglas B. Sigourney is employed by Integrated Statistics.

Douglas B. Sigourney conceived and designed the experiments, performed the experiments, analyzed the data, prepared figures and/or tables, authored or reviewed drafts of the article, and approved the final draft.

Annamaria DeAngelis conceived and designed the experiments, performed the experiments, prepared figures and/or tables, authored or reviewed drafts of the article, annotated and processed acoustic data, and approved the final draft.

Danielle Cholewiak conceived and designed the experiments, authored or reviewed drafts of the article, and approved the final draft.

Debra Palka conceived and designed the experiments, authored or reviewed drafts of the article, and approved the final draft.

The following information was supplied regarding data availability:

Data and code are available at GitHub and Zenodo: https://github.com/NEFSC/READ-PSB-DSIGOURNEY-AcousticDataIntegration.

Douglas Sigourney. (2023). Acoustic Data Integration. https://doi.org/10.5281/zenodo.7566500.

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
