# Peer review of "Combining passive acoustic data from a towed hydrophone array with visual line transect data to estimate abundance and availability bias of sperm whales (Physeter macrocephalus)"

_PeerJ, doi:10.7717/peerj.15850_

## Round 0.1 · original submission · Major Revisions

Both reviewers have included a number of comments that I think will improve the clarity and quality of the manuscript. In particular, please pay attention to the comments from Reviewer 1 (clarifying the likelihood for the integrated model, better explanation of the simulation results).

Reviewer 1 ·

Basic reporting

Overall, the authors provide a detailed summary of the existing literature, with in-depth appendices that include a summary of the key challenges involved with analysing visual line-transect and passive acoustic data that are collected simultaneously.

The paper is comprehensive, which is a strength, but it also means there is quite a bit going on. I struggled a little to follow the manuscript in some places. On some occasions I managed to identify that my confusion was related to errors in mathematical notation or definitions/descriptions that I found were a little misleading. I have provided suggested changes in the "additional comments" section below.

Experimental design

The research question is well-defined and very relevant. The paper is rigorous: the authors propose new statistical methods, apply one of the methods to existing data, and conduct a simulation study to evaluate the performance of the methods proposed.

There was one aspect of the method I didn't manage to grasp. In the paper, the authors advertise their method as an integrated model (IM). IMs involve constructing a likelihood for each data set, and combining them into a single joint likelihood. The individual likelihoods share some parameters and/or latent processes. However, as far as I can tell, the methods proposed here involve separately fitting two different models, one to the visual line transect (VLT) data and one to the passive acoustic montoring (PAM) data, as though these two data sets are independent, and there are no shared parameters that are informed by both data sets. However, in other parts of the manuscript there is a suggestion that the dependence is accounted for somehow (e.g., Line 159).

The joint likelihood for the IM isn't presented in the manuscript: is it just the product of the two submodels, thereby assuming independence between the VLT and PAM data sets? Or is something more sophisticated happening that I have missed? One reason for the lack of independence between the data sets is that an animal detected by the VLT in Bin 2 can't also be detected by the PAM, because it must be on the surface. Is this somehow accounted for in the joint likelihood? Overall, there's almost no detail about the VLT submodel (Line 255-258). I don't think there's any text related to constructing a joint likelihood, and in particular how the joint likelihood respects the lack of independence between the data sets. I think the paper would be improved by including more clarity about the model being fitted, and specifically how the likelihoods for the two models are combined into a joint likelihood, in a way that respects their lack of independence.

As a more minor comment, I wasn't sure why a uniform distribution being used on Line 214 for the time a whale had been in a foraging state prior to entering the detection zone. This seems to directly contradict Eq (5) on Line 214: this equation implies that there is no upper-limit on time spent in the foraging state. Wouldn't the correct distribution depend on alpha_T, and couldn't it be determined analytically without too much trouble? For example, if alpha_T is a large negative number, then we'd expect foraging states to be very short.

Validity of the findings

The simulation study is a strong aspect of this paper, because the data are not simulated under the assumptions of the model being fitted, but rather using alternative models for diving behaviour. The proposed models perform well, which is very reassuring and suggests robustness against the assumptions being made here. If anything, I thought the authors could put a bit more emphasis on this particular strength of their method.

In the real-data example, the CV for N_T is considerably smaller than the CV for N_S, which is suprising, given N_T is a derived parameter that depends on N_S. Any uncertainty in N_S will be propagated through to N_T, in addition to any uncertainty in N_B and N_D. It's mathematically possible for N_T to have the smallest variance (e.g., if there are large negative correlations between the posteriors of N_S and N_B, and N_S and N_D), but it still seems a little strange. This could be a genuine-but-surprising result, although I wonder if it's possible that uncertainty for N_T isn't being calculated correctly for some reason? To verify their approach, it would be helpful if the authors assessed credible interval coverage in their simulation study, which I think this is fairly standard practice for simulation studies testing new statistical methods.

Finally, these models have quite a few parameters, including detection function parameters, (beta0, beta1), dive-cycle transition parameters (alpha0, alpha_T), and so on, but only estimates of abundance are reported. I'd be interested in seeing estimates of other parameters, too, so it might be useful to include them somewhere. If the estimated dive-cycle process is biologically realistic in light of results from other studies, then I think that highlighting this finding would strengthen the paper.

Additional comments

Here are few minor line-by-line comments:

L170: Typo. Should say "states".
L187: What does "it is assigned a 1" mean? It would be helpful to introduce notation for the variable that is being assigned a 1.
L203: I think the z_i, j after the phi should be z_i, j - 1.
L203 and elsehwere: The symbol that should be a "phi" appears to be the empty set symbol in displayed equations.
L211-212: Does it matter that a whale swimming in the same direction as the boat will be in each bin longer than a whale swimming in the opposite direction?
L214: What is a_i, 1? I don't think this piece of notation has been introduced. Should this be Time_i, 1?
L230-243: I found some of the descriptions of parameters here confusing; for example, at first I assumed F_T and S_T were sums over all bins, and only realised I was wrong when I looked at Tables 1 and 2. Also, it took me a whole to figure out that N(CMR)_B is just the same thing as N_B. Is that right?
L255: What is D_T? I don't think this notation has been introduced.
L287: I'm not sure what "we simply multiple by the total number of distance bins" means.
L289: Should this say "transitioning"?
L296: In what ways are the simulated data consistent with the assumptions made by the model, and in what ways are they not? I realise more detail is provided in an appendix, but I would have found it useful to get more of an idea about the simulation setting in the main text.

Reviewer 2 ·

Basic reporting

The authors present a general statistical framework for integrating visual sighting data and passive acoustic monitoring data to estimate marine mammal abundance and availability bias. A complex simulation experiment is described, and a case study is also provided using sperm whales.
The overall reasons and benefits of combining passive acoustic data and visual data is introduced with good references and background information.

The manuscript is organized well given the many pieces that are involved in building the framework. There are 5 documents in the supplementary material that are critical for understanding the overall analysis.

The raw data for the case study does not appear to be shared.

Experimental design

The research is original and fills an important knowledge gap in the field of marine mammal population assessments.

Validity of the findings

No data sets for the case study have been provided.

Conclusions could be clearer in the discussion in terms of where this work fits into the broader field.

Additional comments

Introduction
No changes other than the 2 corrections below. The overall reasons and benefits of combining passive acoustic data and visual data is introduced with good references and background information.
Line 53: Include comma after ‘Worldwide,’
Line 113/114: Check wording in ‘…simultaneously with yet independently…’

Methods
Line 207: Typo, ‘… a function of time where time where…’
Line 218: The detection probability of a whale in a given distance bin was modeled using a logistic function, typically an ‘S’ shaped curve that would indicate an increase in detection probability with increasing distance bins. This does not make sense as the opposite is true, so please provide more details about this model and how it was incorporated into the simulation.
Line 223: Typo, ‘… we us the localized parameter…’
Line 249: Missing comma after ‘dive.’
Line 255: Should DT be FT ?
Line 287: Typo, ‘… simply multiply the ?? by the total… ‘
Line 289: Typo, ‘whales transitioning’
Line 300: Include the reference for DTAGs, which should also be capitalized throughout the paper:
Johnson, M. P., & Tyack, P. L. (2003). A digital acoustic recording tag for measuring the response of wild marine mammals to sound. IEEE journal of oceanic engineering, 28(1), 3-12.
Line 329: Should be ‘farthest,’ not ‘furthest’ as it refers to a physical distance.
Line 349: The forward distance intervals were based on the forward distance of the first recorded click. Appendix 2 describes this more, where the forward distances were derived using the localized position estimate of the whale. However, depending on the bearing of the first click, the depth of the whale, movement of the whale, and the total duration of the event, there are errors associated with a forward distance estimate. Was this considered at all, and if not, how might this type of error affect the analysis? I’m particularly interested in how the distance binning process could accommodate such error, or whether that’s necessary.
Results
Line 387: The final abundance estimates from the Hybrid Method and MRDS analysis were described as similar, which is subjective as the MRDS estimate was 21% higher. Please rephrase to be more specific and objective when describing the results.
Line 388: Please check values in Table 3 as the CV of the Hybrid Method referred to on Line 388 is 50% lower than the MRDS CV in the table, not 57%.

Discussion
The discussion topics did not cover as much as I expected given the complexity of this work. Below are suggestions to help strengthen this section and put it into more context for those that are interested in the application of this statistical framework.
General questions to consider:
- How do you think excluding groups producing codas and slow clicks from the analysis affects abundance estimates?
- What other considerations may be needed for other deep-diving species, such as beaked whales?
- Can you expand upon ways that this analysis could be further developed and applied? One example is mentioned in the Conclusion on Line 500, but it would be helpful to elaborate on that more in the discussion as well.
- What are the remaining biases are there that require future work?
Lines 406-414: The second paragraph contains a restatement of the simulation results. If this is incorporated into the results section, then this paragraph could discuss more about the driving factors of each method’s results and how they could be improved, something that is started on line 414.
Line 435: Why is using only one year of survey data better than using data pooled across years?
Line 419: True sperm whale diving behavior is described to be more complex than the simulated diving behavior accounted for and, therefore, we should interpret simulation results ‘with caution.’ This needs to be discussed further. How cautious should we be? Are there ways to incorporate more complex dive behaviors into the analysis? Why was Watwood et al. 2006 the only study considered when setting up the simulated diving behavior? There are also many other publications that have tagged sperm whales to study their dive behavior, some which are included in this manuscript and supplementary materials. I ask these questions to encourage more discussion of this topic given how important the dive behavior parameter is for the simulation. It would also be good to include this as part of the limitations on line 482.
Line 447: This is confusing. How was the comparison made between the CMR-DS method and Hybrid Method if the CMR-DS Method could not be applied as stated in the previous sentence?
Line 455: Comma after ‘limited’
Line 457: Comma after ‘Method’
Line 464: Comma after ‘method’
Line 480: Comma after ‘distance’
Line 488: Barlow et al. 2021 is not in the reference list.

Conclusion:
The last sentence leaves the reader hanging. Can there be a more definitive statement that places the work performed in this study as a stepping stone for future work instead of deferring to the future work that ‘should’ resolve the remaining biases?


Appendix_S1: List of Challenges
A couple typos and grammar errors were found. Please make those corrections.
Line 3: change ‘ot’ to ‘to’
Line 35: change ‘more shallow’ to ‘shallower’
Line 70: Missing a period after the article title.
In ‘4. Silent States,’ codas and slow clicks are considered ‘silent.’ Could these click types eventually be included?

Appendix_2: Processing and formatting passive acoustic data
1. The steps for processing the passive acoustic data in PAMGuard are explained well. However, could you please clarify what you mean by ‘event?’ I know this is a term used specifically when processing data in PAMGuard, so it would be helpful to explain what exactly that means in terms of single whales, groups of whales, etc.


Appendix_S3: Description of the Zone of Overlap
Line 6: Include comma after (Watwood et al. 2006).
Line 11: Consider rewording this sentence to make it easier to understand, ‘In both scenarios, we assume the PAM platform’s zone of detection is greater than the VLT platform as it extends further in front of and behind the ship.’ Either way, please change ‘further’ to ‘farther.’
Line 12: change ‘then’ to ‘than’
Line 12-13: ‘The zone of detection for the VLT platform is represented by the red box and stops abeam of the ship.’ The red box is then described as the ‘zone of overlap’ in the figure caption. I think it is technically both the detection zone for the VLT platform and overlap, but this should be explained more clearly to avoid confusion.
Line 29: ‘Conversely, whale B transitions out of the foraging state after bin 2…’ The figure shows this transition occurring after bin 3 indicated by the end of the yellow dotted line, please double-check.
Figure S3a & b: Bin labels 5-11 are partially hidden and all are not aligned with ‘Bin 1.’ Please tidy them up.

Appendix_S4: Description of Simulations
Please explain why the ship speed was set to 250 m/min, which equates to ~8 knots. Standard NOAA VLT surveys operate at 10 knots, which would be ~309 m/min. The latter value was used to set the distance bins in the hybrid method (S5), so how comparable are these results to the simulation experiment?

Appendix_S5: Hybrid method
Line 18: Typo in click trains
Line 49: Missing a comma after ‘occur’

Table 3: The caption should specify that these parameter estimates come from the case study.

---

## Round 0.2 · accepted · Accept

Thanks for addressing all of the reviewer comments -- the terminology change (integrated -> combining) is more clear, and I appreciate the additional SI information that has been included.